# Tracheostomy in COVID-19 acute respiratory distress syndrome patients and follow-up: A parisian bicentric retrospective cohort

Diane Evrard[1]*, Igor Jurcisin[2], Maksud Assadi[3], Juliette Patrier[4], Victor Tafani[1], Nicolas Ullmann[5], Jean-François Timsit[4], Philippe Montravers[3], Béatrix Barry[1], Emmanuel Weiss[2,6], Sacha Rozencwajg[3]

1 Department of Otorhinolaryngology, Bichat Hospital, Paris, France, 2 Department of Anesthesiology and Critical Care, Beaujon hospital, DMU Parabol, AP-HP.Nord, Paris, France, 3 Department of Anesthesia and Surgical Intensive Care Unit, Bichat Hospital, Université de Paris, UFR Denis Diderot, INSERM UMR 1152, ANR10-LABX-17, Paris, France, 4 Medical Intensive Care Unit, Bichat Hospital, Paris, France, 5 Department of Oral and Maxillofacial surgery, Beaujon Hospital, Paris, France, 6 Inserm UMR-S1149, Inserm et Université de Paris, Paris, France

* evrard.diane@gmail.com

**Data Availability Statement:** All relevant data are within the manuscript and its Supporting Information files.

## Abstract

### Background

Tracheostomy has been proposed as an option to help organize the healthcare system to face the unprecedented number of patients hospitalized for a COVID-19-related acute respiratory distress syndrome (ARDS) in intensive care units (ICU). It is, however, considered a particularly high-risk procedure for contamination. This paper aims to provide our experience in performing tracheostomies on COVID-19 critically ill patients during the pandemic and its long-term local complications.

### Methods

We performed a retrospective analysis of prospectively collected data of patients tracheostomized for a COVID-19-related ARDS in two university hospitals in the Paris region between January 27th (date of first COVID-19 admission) and May 18th, 2020 (date of last tracheostomy performed). We focused on tracheostomy technique (percutaneous *versus* surgical), timing (early *versus* late) and late complications.

### Results

Forty-eight tracheostomies were performed with an equal division between surgical and percutaneous techniques. There was no difference in patients' characteristics between surgical and percutaneous groups. Tracheostomy was performed after a median of 17 [12–22] days of mechanical ventilation (MV), with 10 patients in the "early" group ($\leq$ day 10) and 38 patients in the "late" group (> day 10). Survivors required MV for a median of 32 [22–41] days and were ultimately decannulated with a median of 21 [15–34] days spent on cannula. Patients in the early group had shorter ICU and hospital stays (respectively 15 [12–19] *versus* 35 [25–47] days; p = 0.002, and 21 [16–28] *versus* 54 [35–72] days; p = 0.002) and

**Funding:** The author(s) received no specific funding for this work.

**Competing interests:** The authors have declared that no competing interests exist.

**Abbreviations:** ARDS, acute respiratory distress syndrome; COVID-19, coronavirus infectious disease 19; ECMO, extracorporeal membrane oxygenation; ETT, endotracheal tube; FFP2, filtering facepiece particles 2; ICU, intensive care unit; MV, mechanical ventilation; PT, percutaneous tracheostomy; ST, surgical tracheostomies.

spent less time on MV (respectively 17 [14–20] and 35 [27–43] days; p<0.001). Interestingly, patients in the percutaneous group had shorter hospital and rehabilitation center stays (respectively 44 [34–81] *versus* 92 [61–118] days; p = 0.012, and 24 [11–38] *versus* 45 [22–71] days; p = 0.045). Of the 30 (67%) patients examined by a head and neck surgeon, 17 (57%) had complications with unilateral laryngeal palsy (n = 5) being the most prevalent.

## Conclusions

Tracheostomy seems to be a safe procedure that could help ICU organization by delegating work to a separate team and favoring patient turnover by allowing faster transfer to step-down units. Following guidelines alone was found sufficient to prevent the risk of aerosolization and contamination of healthcare professionals.

## Introduction

The SARS-CoV-2 pandemic created new challenges for healthcare professionals all over the globe. In particular, intensive care units (ICU) have been overwhelmed with patients suffering from acute respiratory distress syndrome (ARDS), the most severe form of respiratory failure. The understanding of coronavirus infectious disease 19 (COVID-19) related ARDS pathophysiology has led to personalized care for these patients [1].

Among these critically ill patients, some will experience prolonged mechanical ventilation (MV) or difficult weaning. Tracheostomy is often proposed as a weaning strategy for these patients due to its proven benefits: less need for deep sedation, shorter weaning time, and shorter ICU and hospital stay [2]. In the context of the COVID-19 pandemic, tracheostomy (especially early tracheostomy) has also been seen as a good option to optimize the organization of the healthcare system [3] in a context of limited availability of ICU beds and sedative drugs [4]. Nevertheless, tracheostomy is considered a particularly high-risk procedure for the healthcare professionals involved due to droplets and spillage of blood and bronchial secretions. Indeed, the World Health Organization reported an increased risk of contamination for healthcare workers performing tracheostomies [5–7]. Thus, benefits and harms of tracheostomy during the COVID-19 pandemic need to be evaluated.

This paper provides our experience in performing tracheostomies for COVID-19-related ARDS patients during the pandemic. We analyzed tracheostomy techniques, early outcomes and airway complications as well as the serology of staff involved.

## Methods

### Study design

The study was approved by Paris-North Ethically Committee and the French Anesthesiology and Critical Care Medicine Society Ethical Committee, and informed consent was waived as part of a public health outbreak investigation. We performed a retrospective analysis of prospectively collected data from all consecutive patients who underwent a tracheostomy for a COVID-19 respiratory failure at Bichat and Beaujon University Hospitals in the Paris region (France) during the "first wave" between January 27th (date of first COVID-19 admission) and May 18th, 2020 (last tracheostomy performed). The study was approved by the local ethical committee and informed consent was waived as part of a public health outbreak investigation.

## Population

**Patient management in ICU.**   Patient management was discussed daily within the ICU team. It followed current best practice and local guidelines that included regular assessment of sedation, MV parameters and ventilator-associated pneumonia criteria according to guidelines in place [8–10]. COVID-19 specific treatments were only given as part of ongoing concomitant clinical trials.

**Tracheostomy decision and procedure.**   The decision to perform a tracheostomy was taken by a multidisciplinary team and general indications were prolonged MV and difficult weaning [11]. Also, as the pandemic progressed, an increased turnover of patients in ICU was needed to ensure that new patients could be admitted. This objective could only be met by transferring patients recovering from ARDS to "step-down units" (i.e., units which can support tracheostomized patients without any other organ dysfunction). Thus, some tracheostomies were performed based on this particular indication.

As recommended [12–15], surgical tracheostomies (ST) were performed by two senior head and neck surgeons, one intensivist and/or anesthesiologist, two operating theater nurses and one ICU nurse. Percutaneous tracheostomies (PT) were performed by two intensivists and/or anesthesiologists.

To reduce the risk of exposure to the virus, the following rules were applied [12–15]: clamping endotracheal tube (ETT) when the trachea is open and fast insertion of cannula and balloon inflation.

For both, protective apparel included a waterproof cap, goggles with an anti-spitting splash screen, a filtering facepiece particles 2 (FFP2) mask, a disposable waterproof surgical apron, two pairs of surgical gloves, and plastic shoe covers. For all procedures, a portion of the tracheal ring in the shape of a square was excised.

As defined by most studies [16, 17], we considered tracheostomies "early" if they were performed before day 10 from the intubation and "late" otherwise.

## Outcomes

**Patients and procedure.**   Patient data included demographics, date of first COVID-19 symptoms and polymerase chain reaction (PCR) results, level of respiratory support (oxygen therapy, MV support), ICU and hospital outcomes (length of MV, length of stay at hospital, vital status). Patients were also evaluated six months after hospital discharge to evaluate MV and tracheostomy complications as part of a routine post-ICU consultation.

Procedure data included timing of tracheostomy, length of procedure, and complications.

## Healthcare professionals' SARS-CoV-2 status

Healthcare professionals had a PCR nasal swab performed if considered contact cases on clinical suspicion of contamination. Also, as part of an ongoing study looking at healthcare professionals' seroconversion status (SEROCOV NCT04304690), a SARS-CoV-2 serology was performed between two and four weeks after their last tracheostomy.

## Statistical analysis

Data are expressed in median [interquartile range] or percentages as appropriate. Comparison analyses were performed using a Chi-squared test. A p-value below .05 was considered statistically significant. Statistics were performed using Prism 8.0 (GraphPad, La Jolla, USA).

## Results

During the study period, among 1733 patients hospitalized for COVID-19 at Bichat and Beaujon University Hospitals, 300 were hospitalized in ICU, all requiring invasive MV. Forty-eight tracheostomies were performed representing 16% of mechanically ventilated patients with an equal division between surgical and percutaneous techniques. Patients' characteristics are summarized in **Table 1**. Ten tracheostomies were considered early and 38 were considered late (**Fig 1**).

Regarding surgical tracheostomies, 18 (75%) were performed in negative-pressure ICU rooms and 6 (25%) in the operating theater.

### Tracheostomy procedure and complications

Tracheostomy was performed after a median of 17 [12–22] days of MV. The mean duration for the ST procedure was 21 [10–35] minutes. No data was available concerning the duration for the percutaneous tracheostomy (PT) procedure.

Three complications were reported during PT due to technical difficulties, all leading to severe hypoxemia with 2 conversions to a surgical technique. One patient presented a cardiac arrest (no flow null, low flow of 2 minutes) due to hypoxemia, with no neurological consequences. These two patients were included in the ST group.

### Patient outcomes

Patient outcomes are summarized in **Table 2** and **Fig 2**. At 6 months, overall survival was 85%. Five patients died in ICU (1 septic shock due to abdominal abscess, 4 withdrawal of care) and 2 later during hospitalization (1 patient died of a septic shock due to a ventilator-associated pneumonia, the other one after withdrawal of care). The median length of stay in ICU and in hospital for survivors was 31 [18–46] days and 48 [24–61] days respectively. Survivors required MV for 32 [25–41] days and were spontaneously breathing on cannulation for 7 [2–14] days. All survivors were ultimately decannulated with a median of 21 [15–34] days spent on cannula.

### Healthcare professionals' SARS-CoV-2 status

The eight implicated surgeons remained healthy after performing all ST. Their serology blood tests for SARS-CoV-2 tested 3 weeks after the last tracheostomy were all negative. No intensivist and/or anesthesiologist had serology blood conversion due to professional exposure.

### Comparison between early and late tracheostomies

Patients in the two subgroups were similar except for the time from hospitalization to ICU admission (respectively 3 [2–5] versus 1 [0–3] days for the early and late group; p = 0.035), and the need for prone positioning before tracheostomy (respectively 30% versus 78%; p<0.001). Patients in the early group had shorter ICU and hospital stay (respectively 15 [12–19] and 35 [25–47] days; p = 0.002 and 21 [16–28] versus 54 [35–72] days; p = 0.002) and spent less time on MV (respectively 17 [14–20] and 35 [27–43] days; p<0.001) but statistically similar time on cannula (respectively 14 [11–18] and 23 [15–35] days; p = 0.056) (**Table 2**).

### Comparison between surgical and percutaneous techniques

Patients in the two subgroups were similar except for BMI>30 kg/m2 (respectively 15 (63%) for surgical and 6 (25%) for percutaneous tracheostomies; p = 0.02). Patients in the percutaneous group had shorter hospital and rehabilitation center stay (respectively 44 [34–81] versus 92 [61–118] days; p = 0.012 and 24 [11–38] versus 45 [22–71] days; p = 0.045) (**S1 Table**).

**Table 1.  Patients characteristics during intensive care unit hospitalization with early and late tracheostomy.**

| Variables | All patients (N = 48) | Early tracheostomy (n = 10) | Late tracheostomy (N = 38) | p-value |
|---|---|---|---|---|
| **Demographics** | | | | |
| Age, median [IQR]—yr | 56 [47–65] | 52 [48–68] | 57 [46–64] | 1 |
| Male—no. (%) | 36 (75) | 8 (80) | 28 (74) | 0.682 |
| BMI, median [IQR]—kg/m$^2$ | 29.1 [26.7–32.6] | 28.9 [23.4–31.7] | 29.8 [27.7–32.8] | 0.239 |
| BMI > 30 kg/m2—no. (%) | 21 (44) | 3 (30) | 18 (47) | 0.531 |
| Chronic disease—no. (%) | | | | |
| • Chronic heart disease | 6 (13) | 2 (20) | 4 (11) | 0.788 |
| • Chronic kidney disease | 4 (8) | 0 | 4 (11) | 0.668 |
| • Obstructive lung disease[a] | 6 (13) | 0 | 6 (16) | 0.420 |
| • Obstructive sleep apnea | 4 (8) | 0 | 4 (11) | 0.668 |
| • Immunosuppression[b] | 6 (13) | 4 (40) | 2 (5) | 0.016 |
| Pregnancy—no. (%) | 1 (2) | 0 (0) | 1 (3) | 1 |
| Cardiovascular risk factors—no. (%) | | | | |
| • Hypertension | 23 (48) | 2 (20) | 21 (55) | 0.103 |
| • Diabetes mellitus | 14 (29) | 3 (30) | 11 (29) | |
| • Current smoker | 11 (23) | 3 (30) | 8 (21) | 0.860 |
| **COVID-19** | | | | |
| Symptoms to hospital admission, median [IQR]—days | 7 [5–8] | 7 [3–9] | 7 [5–8] | 0.790 |
| Hospital admission to MV, median [IQR]—days | 2 [1–3] | 3 [2–5] | 1 [0–3] | **0.035** |
| **Characteristics in ICU** | | | | |
| Severity scoring | | | | |
| • SAPS 2—mean [range] | *N = 37* | *N = 9* | *N = 28* | |
| | 43 [20–74] | 39 [22–54] | 44 [20–74] | 0.336 |
| • APACHE II—median [IQR] | *N = 40* | *N = 9* | *N = 31* | |
| | 8 [7–11] | 11 [8–17] | 8 [7–11] | 0.080 |
| • SOFA at ICU admission—median [IQR] | *N = 42* | *N = 10* | *N = 32* | |
| | 3 [2–4] | 3 [2–7] | 4 [2–4] | 0.302 |
| First 24 hours of mechanical ventilation | | | | |
| • PaO2/FiO2 ratio—median [IQR] | *N = 42* | *N = 10* | *N = 32* | |
| | 120 [93–139] | 121 [93–145] | 120 [100–136] | 0.608 |
| • PEEP—median [IQR], cmH2O | *N = 42* | *N = 10* | *N = 32* | |
| | 12 [10–12] | 12 [10–12] | 12 [10–14] | 0.402 |
| Mechanical ventilation | | | | |
| • Neuromuscular blockades- no. (%) | 48 (100) | 10 (100) | 38 (100) | 1 |
| • Prone positioning—no. (%) | 32 (67) | 3 (30) | 29 (76) | **0.017** |
| • *If proned, number—median [IQR]* | 3 [2–4] | 1 [1–1.5] | 3 [2–4.5] | **0.007** |
| • ECMO—no. (%) | 6 (13) | 0 | 6 (16) | 0.420 |
| Organ dysfunction during ICU stay | | | | |
| • Vasopressors—no. (%) | 48 (100) | 10 (100) | 38 (100) | 1 |
| • Renal replacement therapy—no. (%) | 14 (29) | 1 (10) | 13 (34) | 0.268 |
| Patients with specific treatments for COVID-19 | 24 (50) | 3 (30) | 21 (55) | 0.286 |
| • Hydroxychloroquine—no. (%) | 1 (2) | 0 | 1 (3) | |
| • Steroids—no. (%) | 14 (29) | 1 (10) | 13 (34) | |
| • Lopinavir/ritonavir—no. (%) | 15 (31) | 1 (10) | 14 (37) | |
| • Anakinra—no. (%) | 2 (4) | 0 | 2 (6) | |
| • Tocilizumab—no. (%) | 2 (4) | 1 (10) | 2 (6) | |
| • Remdesivir—no. (%) | 2 (4) | 1 (10) | 0 | |

*(Continued)*

**Table 1.** (Continued)

| Variables | All patients (N = 48) | Early tracheostomy (n = 10) | Late tracheostomy (N = 38) | p-value |
|---|---|---|---|---|
| Tracheostomy | | | | |
| Technique | | | | 0.722 |
| • Surgical | 24 (50) | 6 (60) | 18 (47) | |
| • Percutaneous | 24 (50) | 4 (40) | 20 (53) | |
| Delay from MV, median [IQR]—days | 17 [12–22] | 8 [6–9] | 19 [14–26] | <0.001 |

<sup>a</sup> obstructive lung disease (n = 6): chronic obstructive pulmonary disease (COPD, n = 3) or asthma (n = 3).

<sup>b</sup> immunosuppression (n = 6): cirrhosis with a CHILD score > B (n = 1), solid organ transplant (n = 3), prolonged corticosteroids (n = 2).

Data are expressed in number (%) or median [interquartile range] as appropriate.

*Abbreviations*: APACHE = Acute Physiology And Chronic Health Evaluation; BMI = body mass index; COVID-19 = coronavirus infectious disease-19; ECMO = extracorporeal membrane oxygenation; FiO2 = inspired fraction of oxygen; ICU = intensive care unit; IQR = interquartile range; MV = mechanical ventilation; PCR = polymerase chain reaction; PaO2 = arterial partial pressure of oxygen; PEEP = positive end-expiratory pressure; SAPS 2 = severity acute physiologic score 2; SOFA = Sequential Organ Failure Assessment.

### Long-term head and neck complications

Thirty patients (63%) were examined by a head and neck senior consultant with a median of 121 [79–147] days after being discharged from ICU. Of the 30 patients examined, 17 (57%) had complications due to prolonged intubation. Unilateral laryngeal palsy was found in 5 (17%) patients. The other complications were dysphonia (30%), laryngeal edema (13%),

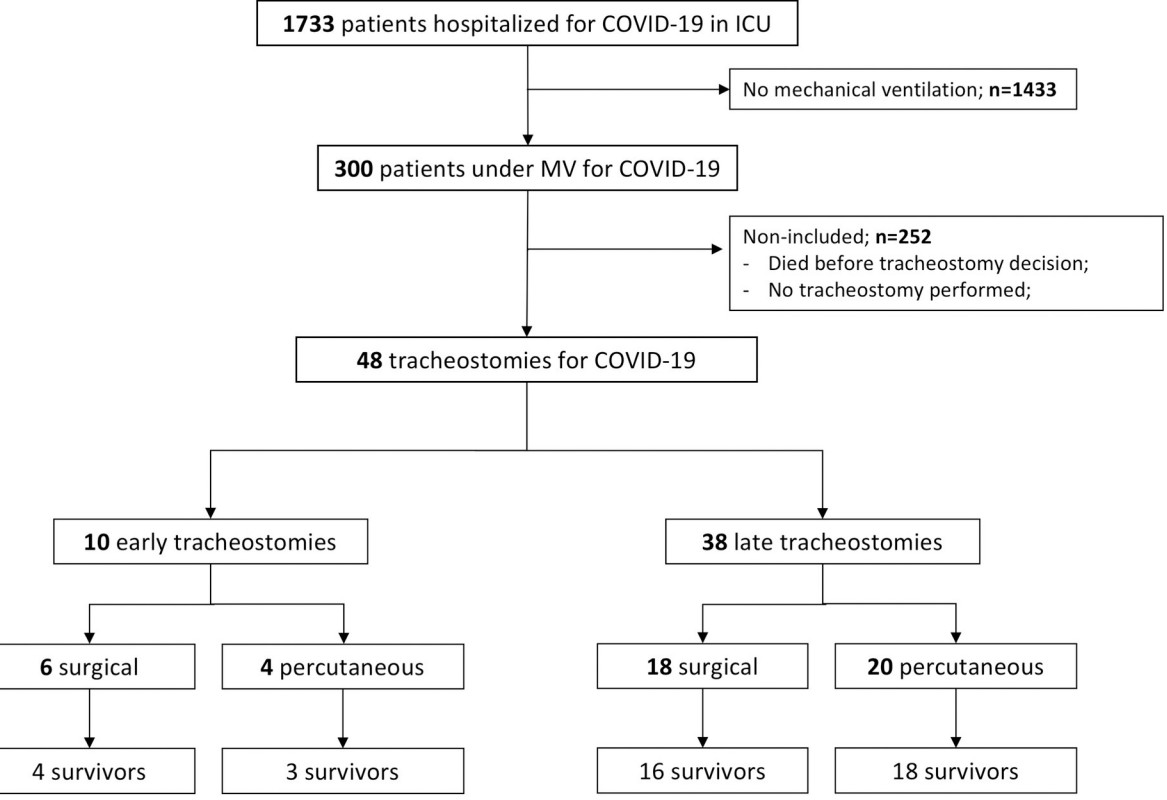

**Fig 1. Flow chart.** *Abbreviations*: COVID-19 = coronavirus infectious disease 19; ICU = intensive care unit.

**Table 2. Patients outcomes after intensive care unit hospitalization with early and late tracheostomy.**

| Outcomes | All patients (N = 48) | Early tracheostomy (n = 10) | Late tracheostomy (N = 38) | p-value |
|---|---|---|---|---|
| Discharge and vital status | | | | |
| Discharged alive from ICU—no. (%) | 43 (90) | 8 (80) | 35 (92) | 0.594 |
| Discharged alive from hospital—no (%) | 41 (85) | 7 (70) | 34 (89) | 0.294 |
| Returned home—no (%) | 41 (85) | 7 (70) | 34 (89) | 0.294 |
| ICU length of stay, days | | | | |
| *All patients* | 32 [18–47] | 14 [12–20] | 38 [25–48] | 0.002 |
| *Surviving patients* | N = 41 | N = 7 | N = 34 | 0.002 |
| | 31 [18–46] | 15 [12–19] | 35 [25–47] | |
| Hospital length of stay, days | | | | |
| *All patients* | 47 [25–61] | 21 [15–31] | 52 [35–70] | 0.002 |
| *Surviving patients* | N = 41 | N = 7 | N = 34 | 0.002 |
| | 45 [26–60] | 21 [16–28] | 54 [35–72] | |
| Time from hospital admission to home return, days | N = 41 | N = 7 | N = 34 | 0.03 |
| | 66 [39–114] | 28 [26–33] | 73 [49–114] | |
| Time spent in rehabilitation center, days | N = 41 | N = 7 | N = 34 | 0.007 |
| | 33 [15–50] | 11 [11–15] | 40 [26–64] | |
| Ventilation and tracheostomy | | | | |
| Duration of mechanical ventilation, days | | | | |
| *All patients* | 32 [22–41] | 17 [14–20] | 35 [27–43] | <0.001 |
| *Surviving patients* | N = 41 | N = 7 | N = 34 | <0.001 |
| | 32 [22–40] | 17 [14–20] | 35 [27–43] | |
| Spontaneous breathing on tracheostomy, days | | | | |
| *All patients* | 12 [7–19] | 9 [7–6] | 13 [8–20] | 0.294 |
| *Surviving patients* | N = 41 | N = 7 | N = 34 | 0.294 |
| | 12 [7–19] | 9 [7–16] | 13 [8–20] | |
| Time on cannula, days | | | | |
| *All patients* | 21 [15–34] | 14 [11–18] | 23 [15–35] | 0.052 |
| *Surviving patients* | N = 41 | N = 7 | N = 34 | 0.056 |
| | 21 [15–34] | 14 [11–18] | 23 [15–35] | |
| Post-tracheostomy complications | | | | |
| At least one complication | N = 30 | N = 7 | N = 23 | 0.923 |
| | 17 (57) | 4 (57) | 13 (57) | |
| *Unilateral laryngeal palsy* | 5 (17) | 2 (29) | 3 (13) | - |
| *Dysphonia* | 9 (30) | 2 (29) | 7 (30) | - |
| *Dysphagia* | 6 (20) | 2 (29) | 4 (17) | - |
| *Laryngeal sensitivity dysfunction* | 3 (10) | 1 (14) | 2 (8,7) | - |
| *Laryngeal edema* | 4 (13) | 1 (14) | 3 (13) | - |
| *Tracheal stenosis* | 2 (6,7) | 2 (29) | 0 (0) | - |

Data are expressed in number (%) or median [interquartile range] as appropriate.

*Abbreviations*: ICU = intensive care unit.

dysphagia (20%), tracheal stenosis (6.7%) and laryngeal sensitivity dysfunction (10%). No significant difference for long-term head and neck complications was found either between early and late tracheostomies or between surgical and percutaneous ones. A vocal fold fat injection was performed 4 months after decannulation for one patient with symptomatic unilateral laryngeal palsy. The mean follow-up of these 30 patients was 277 [42–532] days and no further complication was noted.

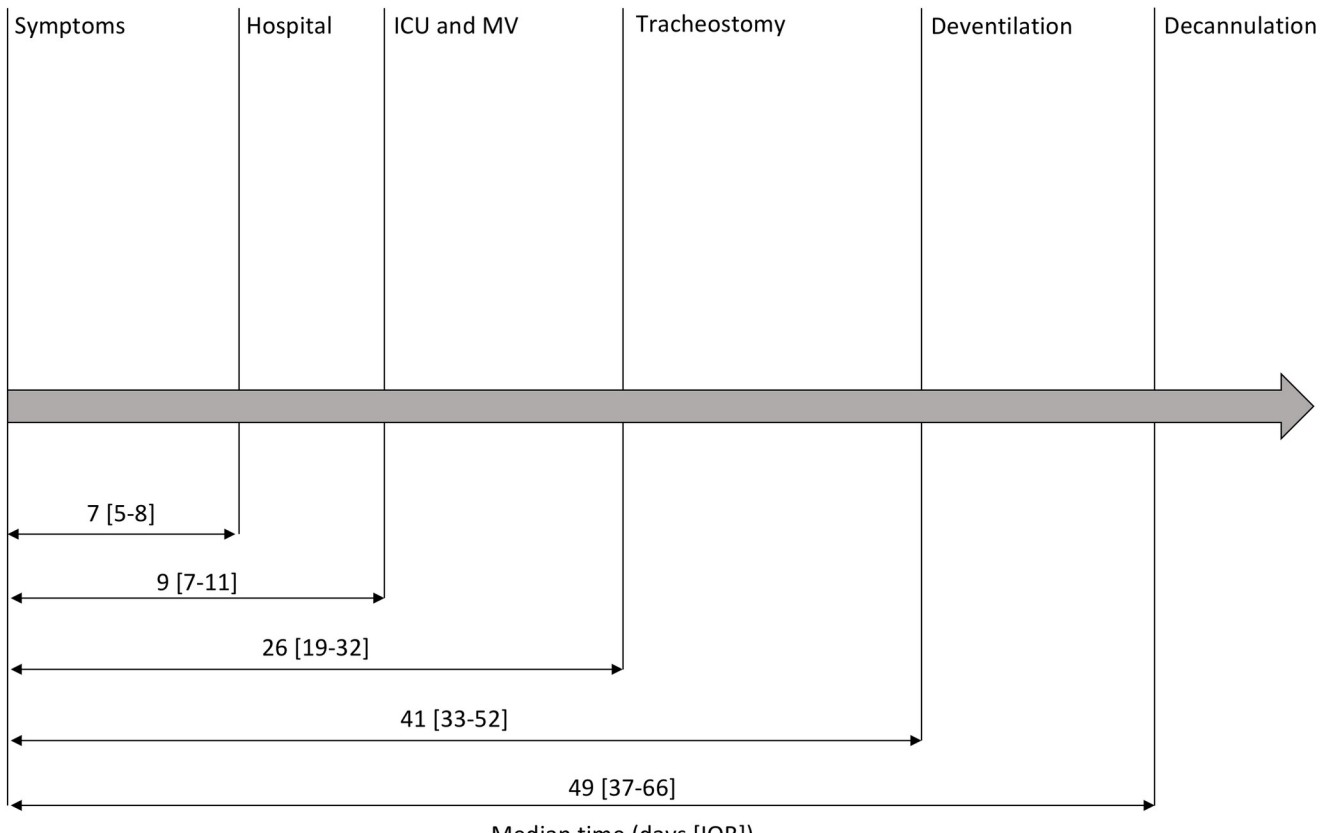

**Fig 2. Timeline of surviving patients tracheostomized after COVID-19 ARDS.** *Abbreviations*: ARDS = acute respiratory distress syndrome; COVID-19 = coronavirus infectious disease 19; ICU = intensive care unit; MV = mechanical ventilation.

## Discussion

How to perform and manage tracheostomy in coronavirus disease (COVID)-19 patients is interesting in terms of procedure, ICU organization and caregivers' protection [13]. In our study, tracheostomy was performed in 16% of our patients. This rate is almost twice higher than in French COVID-ICU study which reported a rate of 9% [18], which might be explained by the high severity of our population. We report 3 (6.3%) complications directly linked to the procedure. The rate of per-procedure complications varied tremendously in published studies ranging from 10.1% [19] to 100% [20]. In a recent meta-analysis [21], Benito *et al.* analyzed 18 studies reporting tracheostomy in COVID-19 patients. Their report showed a large disparity in patient outcomes: weaning from ventilator ranged between 23.3% [22] and 89.3% [23], and successful decannulation between 3.1% [24] and 96.6% [3].

Our study is the first to report long-term complications with a standardized exam by a head and neck senior consultant. We reported a 57% complication rate due to prolonged intubation and/or tracheostomy with no difference depending on timing nor technique used. This study highlights the importance of a head and neck monitoring for these patients. Piazza *et al.* underlined the increase of long-term airways complications due to intubations and tracheostomies for COVID-19 patients [25] and asks for a real "call to action". These complications require specific expertise, must be precociously detected and therefore clinicians must be well aware of those. Our cohort analyzed for the first time these complications with more a year of hindsight.

With the appropriate protection [26], tracheostomy was a safe procedure for the healthcare professionals involved. Similar to previously published studies [3], no contamination was observed after tracheostomy was performed.

Although PT was recommended in ICU by the French National ICU Societies (Société de Réanimation de Langue Française—SRLF and Société Française d'Anesthésie Réanimation—SFAR) experts [11], ST were realized for 50% of patients in our cohort. Particular anatomy of these patients, often with obesity and availability of surgeons could explain this choice.

Timing of tracheostomy in COVID-19 patients remains subject to vivid debate as studies report heterogeneous results [15, 27–30]. In our study with two comparable groups, no difference was observed between the early and late tracheostomies. The debate will most likely be difficult to settle due to the retrospective design of studies performed during the pandemic.

Classic benefits of tracheostomy include reduction of sedation doses, easier mobilization and suction, reduced airway dead space as well as feeding; harms are represented mostly by early and late complications. In the COVID-19 pandemic, potential benefits also included early transfer to step-down units for patients less severe and harms included aerosolization and contamination of healthcare professionals. Due to its retrospective design, our study could shed light on all questions. Nevertheless, several interesting results are worth mentioning.

First, tracheostomy could be performed in these severe critically ill patients without an increased risk of complications. Indeed, the literature reports a global risk of 4.3% for all tracheostomies performed [31].

Second, despite the fact that expert opinions differ on the recommended technique [32], we found no difference between a percutaneous and a surgical approach. For patients with a difficult anatomy or high BMI, we preferred performing a surgical procedure by two senior surgeons in the operating room.

Third, we found that the procedure was safe for healthcare professionals as long as guidelines were respected. Some teams have proposed the use of supplementary protection gear (such as surgical field isolation drape, negative flow hood, snorkeling masks with specific adaptors) [33–35] that may prevent some teams from using the technique. In our study, it seems that standard protective apparel and appropriate rules described by scientific societies are sufficient [14, 15, 36]. Indeed, no surgeons presented any symptoms or a change in SARS-CoV-2 serology 2 weeks after the last procedure, as found in other studies [14, 28, 37].

Fourth, patients who had an early tracheostomy in order to be transferred to step-down units–to help turnover in the context of shortage of ICU beds–showed similar outcomes compared to patients receiving a tracheostomy at a later stage. We even report here that patients had a significantly shorter time in ICU, in hospital and on mechanical ventilation. This is most likely due to the selection of our population who would benefit from an early tracheostomy. Indeed, "late tracheostomies" were performed after a risk-benefits balance had been clearly weighed and patient's risk was considered minimal. On the other side, "early tracheostomies" were performed on patient that were less severe with only 30% proned, none on ECMO, and only one patient had renal replacement therapy. Nevertheless, this is an interesting finding as some authors caution against performing tracheostomy before day 10 of MV [17, 38]. We suggest that timing of tracheostomy should be assessed on a case-by-case basis rather than follow a strict rule. This would allow for more severe patients–who might require additional therapies such as renal replacement therapy or extracorporeal membrane oxygenation (ECMO)–to be admitted to ICU.

Fifth, as ICU staff may be overwhelmed by the amount of work, tracheostomies could be delegated to a separate team. In our case, head and neck surgeons not used to regularly perform percutaneous tracheostomies were trained over a week.

## Strengths and limitations of our study

Our study is the first to report prolonged long-term complications after tracheostomy performed in COVID-19 patients. In a follow-up consultation, local examination highlighted that more than half of the patients presented at least one pharyngo-laryngeal complication. The prevalence of complications is comparable with those reported for non-COVID-19 patients requiring prolonged MV or tracheostomy [39].

Our study suffers some limitations. First, due to its retrospective design and the relatively low number of patients, no causality can be demonstrated. Thus, our conclusions are only associative and need to be taken with precautions. In particular, while duration of MV or length of stay in ICU or hospital was significantly shorter in the early group, this is most likely due to the lower severity of these patients as shown by the lower need for prone positioning. Second, our study took place in two tertiary hospitals, Bichat being an expert center for epidemiological and biological risk, thus generalizability to other settings and hospitals may be limited. Nevertheless, similar results, especially regarding contamination of healthcare professionals, were found in other studies.

Tracheostomy seems to be a safe procedure that could help ICU organization by delegating work to a separate team and favoring patient turnover by allowing faster transfer to step-down units. Following guidelines alone was sufficient to prevent the risk of aerosolization and contamination of healthcare professionals.

## Supporting information

**S1 Table. Patients characteristics during intensive care unit hospitalization and outcomes after intensive care unit hospitalization with surgical and percutaneous tracheostomy.**
*Abbreviations*: BMI = body mass index; COVID-19 = coronavirus infectious disease-19; ECMO = extracorporeal membrane oxygenation; ICU = intensive care unit; IQR = interquartile range; MV = mechanical ventilation; PCR = polymerase chain reaction; PEEP = positive end-expiratory pressure SAPS 2 = severity acute physiologic score 2.
(DOCX)

**S1 Data.**
(XLSX)

## Acknowledgments

We would like to thank Marie Lyager for her English proofreading.

## Author Contributions

**Conceptualization:** Diane Evrard, Victor Tafani.

**Data curation:** Igor Jurcisin, Maksud Assadi, Juliette Patrier, Victor Tafani, Nicolas Ullmann, Sacha Rozencwajg.

**Investigation:** Diane Evrard, Igor Jurcisin, Maksud Assadi, Juliette Patrier, Victor Tafani, Nicolas Ullmann, Sacha Rozencwajg.

**Methodology:** Diane Evrard, Victor Tafani, Sacha Rozencwajg.

**Supervision:** Philippe Montravers, Béatrix Barry, Emmanuel Weiss.

**Validation:** Jean-François Timsit, Philippe Montravers, Béatrix Barry, Emmanuel Weiss.

**Writing – original draft:** Diane Evrard, Sacha Rozencwajg.

**Writing – review & editing:** Diane Evrard.

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
