## [Decision Letter · Decision Letter 0]

23 Aug 2021

PONE-D-21-20812

Tracheostomy in COVID-19 acute respiratory distress syndrome patients and follow-up: a parisian bicentric retrospective cohort

PLOS ONE

Dear Dr. Evrard,

Thank you for submitting your manuscript to PLOS ONE. After careful consideration, we feel that it has merit but does not fully meet PLOS ONE’s publication criteria as it currently stands. Therefore, we invite you to submit a revised version of the manuscript that addresses the points raised during the review process.

Please address the issues and revise accordingly.

We look forward to receiving your revised manuscript.

Kind regards,

Academic Editor

PLOS ONE

4. Please upload a new copy of Figure 1 and Figure 2 as the detail is not clear. Please follow the link for more information: " ext-link-type="uri" xlink:type="simple">https://blogs.plos.org/plos/2019/06/looking-good-tips-for-creating-your-plos-figures-graphics/"
https://blogs.plos.org/plos/2019/06/looking-good-tips-for-creating-your-plos-figures-graphics/

Reviewers' comments:

Reviewer's Responses to Questions

**Comments to the Author**

1. Is the manuscript technically sound, and do the data support the conclusions?

Reviewer #1: Yes

Reviewer #2: Yes

2. Has the statistical analysis been performed appropriately and rigorously? 

Reviewer #1: Yes

Reviewer #2: Yes

3. Have the authors made all data underlying the findings in their manuscript fully available?

Reviewer #1: Yes

Reviewer #2: Yes

4. Is the manuscript presented in an intelligible fashion and written in standard English?

Reviewer #1: Yes

Reviewer #2: Yes

5. Review Comments to the Author

Reviewer #1: Evrard et al. describe their single center experience on tracheostomies in patients with COVID-19 and ARDS. Overall, the manuscript is well-written and the statistics are appropriate.

Major concern:

There are a significant number of publications on outcomes after tracheostomies for COVID-19. While the authors note their longer-term follow-up, I am worried there is somewhat limited new information in this topic. Further discussion into long-term follow-up outcomes, and investigation of associations of complications/long-term effects may be helpful to add new information to existing literature.

Reviewer #2: I read with interest about this study about tracheostomy in COVID-19 patients. Generally, it provided an overview of tracheostomy during early pandemic. Several issues should be further clarified.

Major:

1. In the current study, all surviving patients were weaned from mechanical ventilation and finally decannulated. However, in previous publised studies, mean ventilator weaning rate was less than 60% and mean decannulation rate was less than 40%. How to explain these differences?

2. Several factors were associated with difficult weaning and prolonged decannluation process. The authors should provide more details on the enrolled subjects (Severity of ARDS, severity scoring, such as APACHE II, SOFA, etc, underlying diseases, such as COPD, ILDs, malignancy)

3. Several factors had statistically significant differences between early and late tracheostomy groups. However, they were not mentioned in the result section and the possible mechanism should be addressed in the discussion section.

Minor:

1.

METHODS

"The study was approved by the appropriate Institutional Review Board (IRB), and

written informed consent was obtained from all subjects."

"The study was approved by the local ethical

committee and informed consent was waived as part of a public health outbreak investigation."

It will make readers confusion about whether informed consents were available or not?

2.

RESULTS

During the study period, among 1.733 patients hospitalized for COVID-19 at Bichat and

Beaujon University Hospitals, 300 were hospitalized in ICU, all requiring invasive MV.

→1.733 should be 1733

3.Distant head and neck complications

The term "distant" was not used frequently in medical literature. Please provide a better terminology.

4.

Discussion

Potential benefits and harms of tracheostomy during the COVID-19

pandemic

→ It was not frequent seen that a subheading present in the discussion section. Please integrate it into discussion paragraph.

6. PLOS authors have the option to publish the peer review history of their article (what does this mean?). If published, this will include your full peer review and any attached files.

Reviewer #1: No

Reviewer #2: **Yes: **Wei-Chih Chen

---

## [Author Response · Author response to Decision Letter 0]

12 Oct 2021

Dr Diane EVRARD

Department of Head and Neck Surgery

Bichat University Hospital, Assistance Publique—Hôpitaux de Paris

46 Rue Henri Huchard, 75877 Paris, France, 

Telephone: +33677846800

email: evrard.diane@gmail.com

Professor Emily Chenette, PhD

Editors-in-Chief, PLOS One

6th October 2021,

Dear Professor, 

Please find attached the revised version of our manuscript entitled “Tracheostomy in COVID-19 acute respiratory distress syndrome patients and follow-up: a Parisian bicentric retrospective cohort”, by Evrard et al., that we are revising for publication in PLOS One. 

 We sincerely thank you and the Reviewers for the considerable work incurred reviewing our manuscript. We think that all your constructive comments and suggestions have markedly contributed to improving the quality and readability of our paper. Changes made in the text are in red as instructed.

 We hope that you and the Reviewers find our changes adequate and our paper acceptable for publication. We look forward to hearing from you soon.

 Sincerely yours.

Diane EVRARD on behalf of the authors

 

Reviewer 1 Comments to Author: 

Evrard et al. describe their single-centre experience on tracheostomies in patients with COVID-19 and ARDS. Overall, the manuscript is well-written and the statistics are appropriate.

We would like to thank Reviewer 1 for their appreciation of our work. 

Major concern:

There are a significant number of publications on outcomes after tracheostomies for COVID-19. While the authors note their longer-term follow-up, I am worried there is somewhat limited new information in this topic. Further discussion into long-term follow-up outcomes, and investigation of associations of complications/long-term effects may be helpful to add new information to existing literature.

We agree with Reviewer #1 that several publications have become available regarding tracheostomies in patients with COVID-19 and that our results are concordant with most of the literature published on this topic.

Nevertheless, we believe two things are important and should be noted: 

1) We believe it is important for the readers to obtain more data regarding tracheostomies in patients with COVID-19 as this is most likely to be correct for any upcoming respiratory pandemic and therefore even though our study doesn’t bring any novelty in this field, it may be of importance to the readers;

2) We bring novelty by having reported long-term outcomes specifically linked to tracheostomies in this specific population, and we believe it is major for healthcare professionals to be aware of these complications, especially for those who are not used to this specific literature, not reported elsewhere. Piazza et al. underlined the increase of long-term airways complications due to intubations and tracheostomies for COVID-19 patients and asks for a real “call to action”. 

We have emphasized this in the discussion (Page 14) and hope that those arguments would make Reviewer #1 consider favorably our manuscript for PLOS One.

 

Reviewer 2 Comments to Author: 

I read with interest about this study about tracheostomy in COVID-19 patients. Generally, it provided an overview of tracheostomy during early pandemic. Several issues should be further clarified.

We would like to thank Reviewer 2 for their appreciation of our work. 

Major:

In the current study, all surviving patients were weaned from mechanical ventilation and finally decannulated. However, in previous published studies, mean ventilator weaning rate was less than 60% and mean decannulation rate was less than 40%. How to explain these differences?

As we indicated in the manuscript, Benito et al performed a review of tracheostomy for COVID-19 patients and reported a weaning rate up to 89.3% and a decannulation rate up to 96.6%. These high rates corresponded to Williamson et al. study in which the population of 29 patients was similar to our cohort. 

We believe that these differences may be explained by several factors in our study:

1. For “late tracheostomies”. These late procedures are performed in our centers after a risk-benefits balance is weighed and the patient’s risk of death is considered minimal. Thus - because of our habits - “risk of weaning” and “late tracheostomy” could almost be statistically considered as competing factors, which may explain the absence of death in the late tracheostomy group. 

2. For “early tracheostomies”. 

a. As mentioned in the manuscript, patients eligible for an early tracheostomy were less severe: only 30% were proned, none were on ECMO, only 1 patient had RRT (even if ECMO and RRT are not statistically different due to our limited number of patients). Therefore, we believe the overall good prognosis of our patients is explained by the selection of patients eligible for an early tracheostomy. 

b. Also, our study takes place during the “first wave” during which it is now known that intubation was probably “overused” at the initial part of the disease even if patients did meet ARDS criteria. Nevertheless, we believe this explanation does not totally fit our population as driving pressure at intubation was relatively low - corresponding to a “type L” phenotype - and comparable with the late tracheostomy group.

2. Several factors were associated with difficult weaning and prolonged decannulation process. The authors should provide more details on the enrolled subjects (Severity of ARDS, severity scoring, such as APACHE II, SOFA, etc, underlying diseases, such as COPD, ILDs, malignancy)

As requested, we have added details in enrolled subjects: 

- Severity of ARDS: we added the respiratory parameters during the first 24 hours of ventilation (PF ratio and PEEP) as well as the number of prone positioning;

- Severity scoring: we added the APACHE II and SOFA at ICU admission;

- Underlying diseases: 

- COPD: there was a typo in the table as “COPD” and “asthma” were regrouped under the term “asthma”. We now separated them and provided details under Table 1 “a”;

- ILDS: none of our patients had chronic interstitial lung disease

- immunosuppression: we detailed it under Table 1 “b”.

3. Several factors had statistically significant differences between early and late tracheostomy groups. However, they were not mentioned in the result section and the possible mechanism should be addressed in the discussion section.

Regarding the statistical differences between early and late, all of them are mentioned in the results section:

- characteristics: difference for “hospital admission to MV” and for “prone positioning” reported in the paragraph Comparison between early and late tracheostomies

- outcomes: differences in length of stay in ICU or at hospital and duration of MV which are reported in the same paragraph.

We agree this was not mentioned in the discussion paragraph, we therefore added the following sentence (Page 16) : “We even report here that patients had a significantly shorter time in ICU, in hospital and on mechanical ventilation. This is most likely due to the selection of our population who would benefit from an early tracheostomy. Indeed, “late tracheostomies” were performed after a risk-benefits balance had been clearly weighed and the patient's risk was considered minimal. On the other side, “early tracheostomies” were performed on patient that were less severe with only 30% proned, none on ECMO, and only one patient had renal replacement therapy.” 

Minor:

1. METHODS

"The study was approved by the appropriate Institutional Review Board (IRB), and

written informed consent was obtained from all subjects."

"The study was approved by the local ethical committee and informed consent was waived as part of a public health outbreak investigation."

It will make readers confused about whether informed consents were available or not?

We agree with Reviewer #2 that this paragraph induces confusion. 

We have obtained an IRB approval to collect all data on COVID-19 patients and consent was waived for this study.

We, therefore, deleted the “and written informed consent was obtained from all subjects” and regrouped the two paragraphs into one.

2.RESULTS

During the study period, among 1.733 patients hospitalized for COVID-19 at Bichat and

Beaujon University Hospitals, 300 were hospitalized in ICU, all requiring invasive MV.

→1.733 should be 1733

The typo has been corrected.

3.Distant head and neck complications

The term "distant" was not used frequently in the medical literature. Please provide better terminology.

We have replaced the term “distant” with “long-term”. Please excuse our “Frenchism”.

4. DISCUSSION. 

Potential benefits and harms of tracheostomy during the COVID-19 pandemic

→ It is not frequently seen that a subheading is present in the discussion section. Please integrate it into the discussion paragraph.

We have deleted the subheading into the discussion paragraph as requested.

 

Editors Comments to Author: 

We have corrected the references style.

We have added our data table.

We have deleted this sentence.

4. Please upload a new copy of Figure 1 and Figure 2 as the detail is not clear. Please follow the link for more information: https://blogs.plos.org/plos/2019/06/looking-good-tips-for-creating-your-plos-figures-graphics/" https://blogs.plos.org/plos/2019/06/looking-good-tips-for-creating-your-plos-figures-graphics/

We have increased the dpi of each figure.

---

## [Decision Letter · Decision Letter 1]

23 Nov 2021

Tracheostomy in COVID-19 acute respiratory distress syndrome patients and follow-up: a parisian bicentric retrospective cohort

PONE-D-21-20812R1

Dear Dr. Evrard,

We’re pleased to inform you that your manuscript has been judged scientifically suitable for publication and will be formally accepted for publication once it meets all outstanding technical requirements.

Kind regards,

Sebastian Shepherd

Associate Editor

PLOS ONE

Additional Editor Comments (optional):

Reviewers' comments:

Reviewer's Responses to Questions

**Comments to the Author**

1. If the authors have adequately addressed your comments raised in a previous round of review and you feel that this manuscript is now acceptable for publication, you may indicate that here to bypass the “Comments to the Author” section, enter your conflict of interest statement in the “Confidential to Editor” section, and submit your "Accept" recommendation.

Reviewer #1: All comments have been addressed

Reviewer #2: All comments have been addressed

2. Is the manuscript technically sound, and do the data support the conclusions?

Reviewer #1: Yes

Reviewer #2: Yes

3. Has the statistical analysis been performed appropriately and rigorously? 

Reviewer #1: Yes

Reviewer #2: Yes

4. Have the authors made all data underlying the findings in their manuscript fully available?

Reviewer #1: Yes

Reviewer #2: Yes

5. Is the manuscript presented in an intelligible fashion and written in standard English?

Reviewer #1: Yes

Reviewer #2: Yes

6. Review Comments to the Author

Reviewer #1: The authors have improved their manuscript and addressed my comments. Overall additional long-term information on tracheostomies in COVID patients can be of value.

Reviewer #2: The authors addressed reviewers' and editor's comment adequately. I suggested the manuscript accepted for publication.

7. PLOS authors have the option to publish the peer review history of their article (what does this mean?). If published, this will include your full peer review and any attached files.

Reviewer #1: No

Reviewer #2: No

---

## [Editor Report · Acceptance letter]

9 Dec 2021

PONE-D-21-20812R1 

Tracheostomy in COVID-19 acute respiratory distress syndrome patients and follow-up: a parisian bicentric retrospective cohort 

Dear Dr. Evrard:

I'm pleased to inform you that your manuscript has been deemed suitable for publication in PLOS ONE. Congratulations! Your manuscript is now with our production department. 

Kind regards, 

on behalf of

Dr Sebastian Shepherd 

Staff Editor

PLOS ONE